

# Hybrid identification for *Glycine max and Glycine soja* with SSR markers and analysis of salt tolerance

Fayuan Li[1],[*], Xun Liu[1],[*], Shengyan Wu[1], Qingyun Luo[2] and Bingjun Yu[1]

[1] College of Life Sciences, Nanjing Agricultural University, Nanjing, China
[2] College of Horticulture Sciences, Nanjing Agricultural University, Nanjing, China
[*] These authors contributed equally to this work.

## ABSTRACT

*Glycine max* cultivars Lee68, Nannong 1138-2, and Nannong 8831 were used as the female parents, and hybrid lines ($F_5$) 4,111, 4,076 (N23674 × BB52), 3,060 (Lee68 × N23227), and 185 (Jackson × BB52) that selected for salt tolerance generation by generation from the cross combination of *G. max* and *G. soja* were used as the male parents, 11 (A–K) backcrosses or three-way crosses were designed and 213 single hybrids were harvested. The optimized soybean simple sequence repeat (SSR)–polymerase chain reaction (PCR) system was used to analyze the SSR polymorphism of above parental lines and get the parental co-dominant SSR markers for hybrid identification, and in which 30 true hybrids were gained. The true hybrids (G1, G3, G9, G12, G13, G16) of G cross combination were chosen as the representative for the salt tolerance test, and the results showed that, as exposed to salt stress, the seedlings of G9 line displayed higher salt tolerant coefficient, relative growth rate, and dry matter accumulation, when compared with their female parent Nannong 1138-2, and even performed equally strong salt tolerance as the male parent 3,060. It provides a feasible method of the combination of molecular SSR markers and simple physiological parameters to identify the true hybrids of *G. max* and *G. soja*, and to innovate the salt-tolerant soybean germplasms.

# INTRODUCTION

Salinity is one of the major abiotic stresses that adversely affect crop productivity and quality (*Chinnusamy, Jagendorf & Zhu, 2005*; *Shrivastava & Kumar, 2015*). Approximately 22% of the world's agricultural lands are affected by salinity (*Bhatnagar-Mathur, Vadez & Sharma, 2008*; *Machado & Serralheiro, 2017*). It is reported that more than 800 million hectares of land throughout the world, which account for more than 6% of the world's total land areas, have been reported to be affected by salinity (*Munns & Tester, 2008*). In China, there are about 36 million hectares saline lands. At present, what is worthy to be highly worried is that, the saline lands are expanding with the industry modernization, the increase of irrigated agrarian lands, and greenhouse for vegetables and flowers in agriculture (*Zhang et al., 2011*). Therefore, the genetic improvement of crop salt

Corresponding author
Bingjun Yu, bjyu@njau.edu.cn

tolerance, bioavailability, and development of saline soil, have become the most important approaches in the future agriculture development.

Soybean is one of the main crops in China, and is also the main source of plant oil and protein in the world (*Zhang et al., 2011*; *Adisa & Balogun, 2013*; *Kim, Hwang & Lee, 2017*). It includes the cultivated soybean (*Glycine max* (L.) Merr.) and wild soybean (*G. soja* Sieb. & Zucc.) *G. soja* is known as the ancestor of *G. max* (*Hyten et al., 2006*), and generally, *G. soja* is more salt tolerant than *G. max* (*Luo, Yu & Liu, 2005*; *Patil et al., 2016*). Although the cultivated soybean belongs to the moderate salt-tolerant plant (*Tuyen, Lal & Xu, 2010*), the great achievements in conventional breeding among *G. max* cultivars for improving its salt tolerance are difficult to obtain due to its narrowing basis of genetic germplasms and the relative limited salt tolerance, as well as the lower productivity and economic value in *G. soja* (*Zhang et al., 2011*). One of the effective ways to improve soybean salt tolerance and breed new soybean varieties is to select generation by generation for the hybrids with salt tolerance of heterosis from the cross combination of *G. max* and *G. soja* (*Lee et al., 2004*, *2009*). The soybean hybrids with salt tolerance of heterosis often possess some adverse agronomic characters, such as plant climbing and vining, and small seeds, which can be improved by further backcrossing method to obtain the agricultural cultivars with specific individual target agronomic traits (*Wang et al., 2003*). However, the authenticity of hybrid identification is a very important process in the cross breeding of soybean cultivars. So an accurate, simple, and fast method for identifying the soybean hybrid authenticity is urgently needed.

Molecular markers are powerful genomic tools for increasing the efficiency and precision of breeding practices for crop improvement (*Ashraf & Foolad, 2013*). For example, simple sequence repeat or microsatellite (SSR) markers are widely applied in molecular and genetic map construction, genetic purity identification, genotype fingerprinting, analysis of germplasms diversity, and utilization of heterosis, which with some merits such as simplicity, quickness, rich polymorphism, high stability, and co-dominance (*Burnham et al., 2002*; *Kuroda et al., 2009*; *Patil et al., 2016*). The polymorphic SSR markers, as an ideal molecular technology, has been used widely to identify the variety identification or hybrid purity in rice, wheat, common bean, cotton, rape, peanut, apple, orange, and psidium (*Gaitán-Solís et al., 2002*; *Nandakumar et al., 2004*; *Dawson et al., 2013*; *Tuler et al., 2015*). The construction and optimization of the SSR reaction system and the selection of SSR primers are the important foundation for its application. It was found for the rice cross-breeding that only one pair of primers would be sufficient to identify the hybrids, as long as the distinguished co-dominant SSR markers between the parents were screened clearly (*Yashitola et al., 2002*; *Nandakumar et al., 2004*). In this work, the hybrid lines ($F_5$) 4,076, 4,111, 3,060, and 185 were used as the male parents, and the *G. max* cultivars Lee68, Nannong 1138-2, and Nannong 8831 as the female parents, 11 (A–K) backcrosses or three-way crosses were designed and the harvested hybrids were identified by SSR markers. Then, the salt tolerance of the true hybrid lines of one representative cross combination (G) was evaluated as compared with their parents. We aimed at providing an important theoretical principle for innovation of the salt tolerant soybean germplasms and improvement of salt tolerance in *G. max* by using *G. soja*.

**Table 1 The 11 soybean cross combinations used in this study.**

| Soybean cross combinations | Female × Male |
|---|---|
| A | Lee68 × 4,076 |
| B | Lee68 × 4,111 |
| C | Lee68 × 3,060 |
| D | Lee68 × 185 |
| E | Nannong 1138-2 × 4,076 |
| F | Nannong 1138-2 × 4,111 |
| G | Nannong 1138-2 × 3,060 |
| H | Nannong 1138-2 × 185 |
| I | Nannong 8831 × 4,076 |
| J | Nannong 8831 × 4,111 |
| K | Nannong 8831 × 3,060 |

## MATERIALS AND METHODS

### Plant materials

The hybrid lines ($F_5$) 4,111, 4,076, 3,060, and 185, respectively, selected and gained for salt tolerance generation by generation from the cross combination of *G. max* and *G. soja* (N23674 × BB52; *Zhang et al., 2011*), (Lee68 × N23227; *Li et al., 2012*), and (Jackson × BB52; *Wu & Yu, 2009*), were used as the male parents, and the *G. max* cultivars Lee68 (the salt tolerant, USA), Nannong 1138-2 and Nannong 8831 (with fine agronomic traits, Nanjing, China) were used as the female parents, 11 combinations of backcross or three-way cross were designed (Table 1). The hybrid seeds of 12–28 plants (213 in total) of each combination were harvested individually after they ripened. The suitable amount of seed of each single plant was cultivated in the greenhouse for the next experiments.

### Total leaf DNA extraction

Total genomic DNA was extracted from the young leaves of single plant seedling using cetyltrimethylammonium bromide method (*Siew et al., 2018*).

### PCR amplification and product electrophoresis

All SSR–polymerase chain reaction (PCR) were performed on PCR system (Bio-Rad, Hercules, CA, USA). PCR was performed in a 10 µL reaction volume, its reaction procedure is as follows: 94 °C pre-denaturation 5 min, 94 °C denaturation 30 s, 55 °C annealing 40 s, 72 °C extension 30 s, 30 cycles; 72 °C extension 10 min, then 4 °C insulation (using the most suitable annealing temperature depending on the primer). Amplification products were analyzed by electrophoresis in 8.0% (w/v) denaturing polyacrylamide gel in 1× TBE buffer for 1 h on the DYY-6C electrophoresis apparatus (Beijing Liuyi Factory, Beijing, China) under 220 V constant voltage. Fragments were then visualized by silver staining (Silver sequence staining reagents, Promega, Madison, WI, USA) and sized with 50 base pairs DNA ladder marker (TianGen Biotech Company, Beijing, China). SSR primer sequences (Table S1) were obtained from the soybean public databases (http://soybase.org/resources/ssr.php).

**Table 2 Results of molecular appraisal of 213 single hybrid seedlings of different soybean cross combinations with SSR marker Satt682.**

| Cross combinations | Total number of hybrids | Number of pure hybrids | Proportion of pure hybrids (%) |
|---|---|---|---|
| A | 28 | 8 (A3, A7, A10, A16, A20, A23, A25, A26) | 28.6 |
| B | 14 | 4 (B6, B7, B8, B9) | 28.6 |
| C | 19 | 7 (C2, C3, C9, C10, C12, C13, C14) | 36.8 |
| D | 12 | 0 | 0 |
| E | 20 | 0 | 0 |
| F | 20 | 0 | 0 |
| G | 20 | 6 (G1, G3, G9, G12, G13, G16) | 30.0 |
| H | 20 | 0 | 0 |
| I | 20 | 1 (I9) | 5.0 |
| J | 20 | 0 | 0 |
| K | 20 | 4 (K13, K14, K15, K16) | 20.0 |
| Total | 213 | 30 | 14.1 |

## SSR–PCR reaction system and screening polymorphic primers among parents of different cross combinations

Using L16 ($4^5$) orthogonal design (*Yang, Mu & Wang, 2007*), the exploration orthogonal test was performed at five factors (DNA template concentration, dNTP, primer concentration, Taq enzyme concentration, and $Mg^{2+}$ concentration) and four levels (Table S2). In addition to the table variables, each tube also contained 1.5 µL 10× buffer, primer Satt242 and DNA template. Using the well-established best SSR–PCR reaction system, 24 pairs of primers of SSR markers were selected randomly on 20 linkage groups in soybean (Satt519, Satt467, Satt474, Sat-367, Sat-311, Satt432, Satt444, Satt168, Satt726, Satt161, Satt682, Satt-153, Satt-264, Satt556, Satt194, Satt-207, Satt286, Sat-332, Satt254, Satt147, Satt447, Satt-220, Satt708, Satt368, respectively) for PCR experiment in the genomic DNA of the experimental soybean materials to verify the stability of the obtained system. Then, the randomly selected 18 pairs of SSR primers (Satt682, Satt70, Satt368, Satt440, Sat-246, Sat-240, Sat-262, Satt649, Satt170, Satt242, Satt152, Satt102, Sat-359, Sat-276, Sat-393, Satt530, Satt348, Satt072, respectively) to screen polymorphic primers between parents of different cross combinations, and the markers with clear and stable amplification type, polymorphism, obvious main band, and less shadow tag were selected for the hybrid identification.

## Identification of true soybean backcross or three-way cross hybrid lines

Identification of true hybrids was performed as previously described (*Nandakumar et al., 2004*; *Ben Romdhane et al., 2018*). If the sample of hybrids containing two parental co-dominant bands, could be judged as the true hybrid, otherwise could be false. One polymorphic SSR marker (Satt682) was used in the current study to identify the two parents and 213 generated hybrids (Table 2).

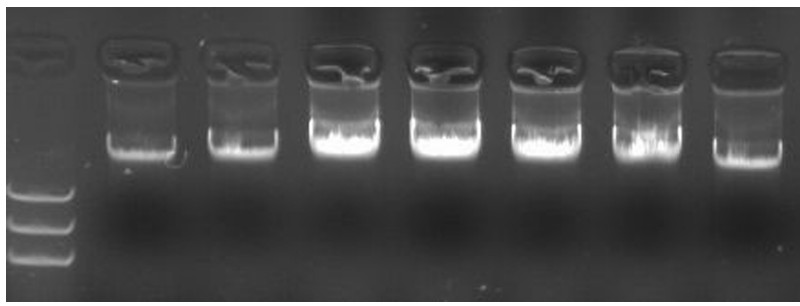

**Figure 1 DNA extracted from leaves of soybean seedlings.** Note: 1–7 DNA extracted from leaves (left to right) of parent Lee68, Nannong 1138-2, Nannong 8831, 4,076 ($F_5$), 4,111 ($F_5$), 3,060 ($F_5$), 185 ($F_5$); M: DNA marker.

## Determination of the salt tolerant coefficient

Salt tolerant coefficient was measured according to the method of *Li et al. (2012)*. When the first pair of unifoliolate leaf of true hybrid lines of cross combination G expanded fully, the seedlings were treated with 50 mmol/L NaCl solution (prepared with 1/2 Hoagland solution) for 5 days, then increased by 25 mmol/L NaCl solution every 5 days till its final concentration increase to 150 mmol/L. All the solutions listed above were replaced every 2 days. The number of the plants appearing leaf salt injury symptom was recorded everyday until all the plants appeared. Salt tolerant coefficient was calculated by the days when the leaf salt injury symptom appeared on the first plant, 50% plants, and 100% plants, respectively.

## Determination of the relative growth rate and dry matter accumulation

According to *Du & Yu (2010)*, when the first pair of unifoliolate leaves expanded fully, the seedlings were treated with 1/2 Hoagland solution (Control) and 120 mmol/L NaCl solution (prepared with 1/2 Hoagland solution) for 10 days. All the solutions listed above were replaced every 2 days. A total of 12 young seedlings were sampled, respectively, for control and salt treatment, and the plant height was measured. Relative growth rate = [(Plant height after treatment − Plant height before treatment)/(Plant height for control − Plant height before treatment)] × 100%. Then the material was fully rinsed in distilled water, dried to constant weight at 80 °C after 105 °C fixing for 10 min. Dry matter accumulation = (Dry weight of salt − Stressed single plant/Dry weight of control single plant) × 100%.

# RESULTS

## DNA extraction and construction of soybean SSR–PCR reaction system

The results of DNA extraction of the experimental materials were displayed with better DNA quality for following SSR–PCR analysis (Fig. 1). The amplification effects were obviously different in the orthogonal designed SSR–PCR reaction system (16 designs) (Fig. 2; Table S2). The clear amplified bands were displayed in the designs No. 1, 2, 6, 7, 10, 13, and 14, but designs No. 3, 8, and 11 showed weak and small amount of amplified

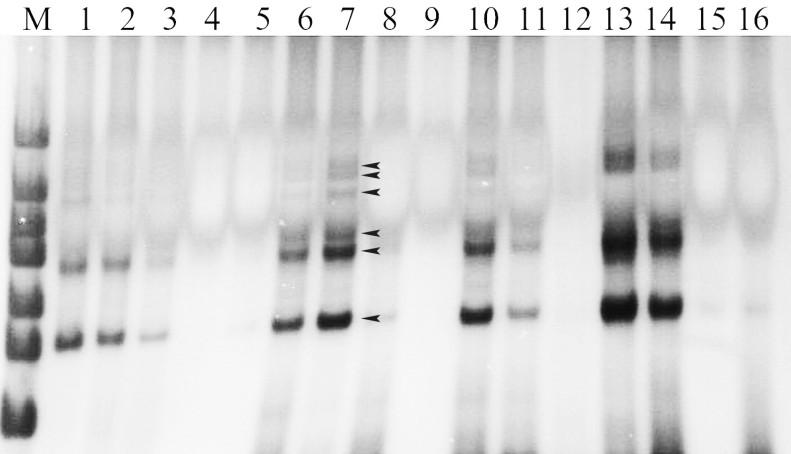

**Figure 2** **The amplified results of 16 designs according to the orthogonal design scheme.** Note: 1–16, designs No. 1–16 denoted in Table S2; M. DNA marker; the used primer is Satt242. Observed bands in lane 7 were marked with arrows.               

bands, designs No. 4, 5, 9, 12, 15, and 16 exhibited no bands. In this study, we selected the design No. 7 as the more appropriate one, for which 10 μL reaction volume containing template DNA 30 ng, dNTP 240 μmol/L, $Mg^{2+}$ 2.0 mmol/L, SSR primer 1.5 μmol/L, Taq enzyme 0.5 U. Then, the randomly selected 24 pairs of primers were used to verify the stability of the system, and showed that all the primers could be all amplified with polymorphic bands (Fig. S1), indicating that this system could be used for soybean genetic diversity of SSR markers.

### True hybrid identification of different soybean cross
In this study, 18 pairs of SSR primers were randomly selected in the soybean 20 linkage groups to screen the polymorphic SSR markers among seven parents of 11 combinations of backcrosses or three-way crosses (A–K). The numbers of parental polymorphic SSR primers varied between 5 and 11 (Table S3), and two pairs of SSR primers (Satt682 and Satt440) were located in $C_1$ and I linkage groups, displayed co-dominant and parental polymorphism among all the parents (Fig. S2). One of them, Satt682 was adopted to identify the true hybrids. For example, in Fig. 3A, it showed that band 3 was the characteristic one of the female parent ($P_1$) and band 4 for the male parent ($P_2$), with regard to the 26th plant, band 1 came from $P_1$, band 2 from $P_2$, and so we can determine that the 26th plant is a true hybrid. Accordingly, 30 true hybrids (accounting for 14.1%) were gained successfully from the harvested 213 single hybrid plants of 11 cross combinations, and partial combinations, such as A, B, C, and G were displayed in Fig. 3.

### Effects of salt stress on seedling salt tolerant coefficient and growth of the true hybrid lines and their parents of cross combination G
The gained-above six true hybrid lines of cross combination G (G1, G3, G9, G12, G13, and G16) and their parents (Nannong 1138-2 and 3,060) were used as the representative experimental materials to evaluate the salt tolerance. When the first plants, 50% or 100%

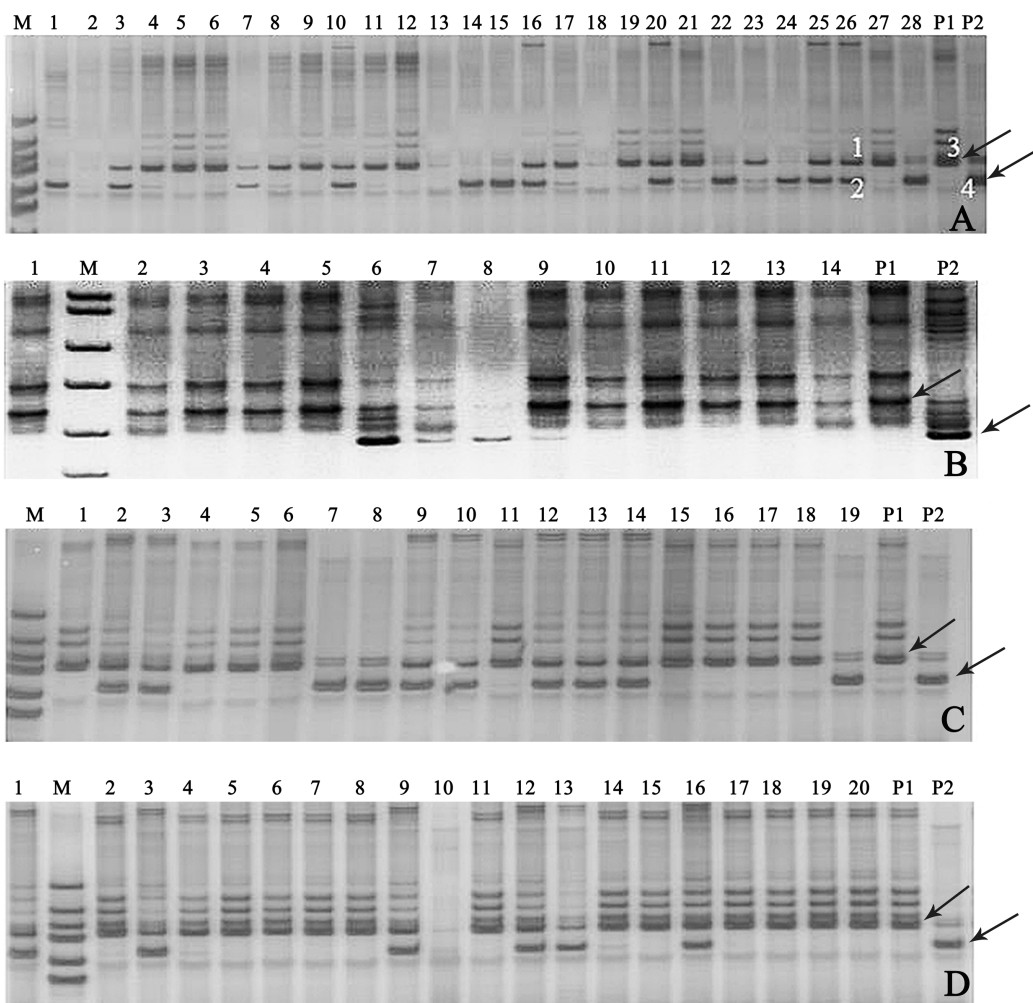

**Figure 3 Gel electrophoresis assay of soybean hybrid lines and their parents using SSR marker Satt682.** Note: (A) Gel electrophoresis of combination A, (B) gel electrophoresis of combination B, (C) gel electrophoresis of combination C, (D) gel electrophoresis of combination G. The numbers are for the samples of the corresponding cross combination lines. $P_1$: the corresponding combination female parents, $P_2$: the corresponding combination male parents, M: DNA marker; the labeled 1, 2, 3, and 4 in (A) amplified, respectively, the bands of the 26th plant and its parents. Observed bands from the parental lines were marked with arrows.                

plants showed leaf salt injury symptoms, the salt tolerant coefficient of female parent Nannong 1138-2 was obviously lower than that of the male parent 3,060. The salt tolerant coefficient of hybrid lines G1, G12, G13, and G16 was almost equal to Nannong 1138-2, but significantly lower than 3,060 when 50% or 100% plants showed leaf salt injury symptoms. However, the salt tolerant coefficient of hybrid lines G3 and G9, was roughly equal to the male parent 3,060 at the above three different salt injury levels, and thus displayed their stronger salt tolerance, especially for G9 (Fig. 4).

Under 120 mmol/L NaCl for 10 days, the relative growth rate of seedlings of the above true hybrid lines and their parents of cross combination G all decreased evidently when compared with the control, the declines of hybrid lines G3 and G16 (33.3% and 35.5%, respectively) were equal to their male parent 3,060 (34.4%), but were significantly less than

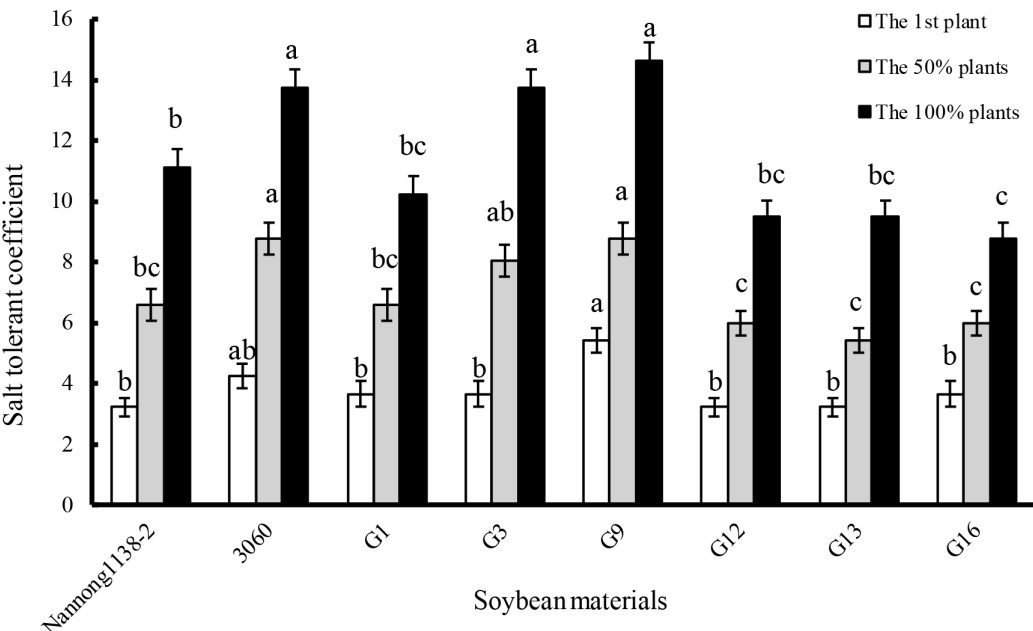

**Figure 4 Comparison of salt tolerant coefficient in the hybrid lines and their parents of cross combination G.** Note: The first plant: salt tolerant coefficient was calculated when the leaf salt injury symptom appeared on the first plant seedling; the 50% plants: salt tolerant coefficient was calculated when the leaf salt injury symptom appeared on the 50% plant seedlings; the 100% plants: salt tolerant coefficient was calculated when the leaf salt injury symptom appeared on the 100% plant seedlings. The different lower case letters indicate significance ($p < 0.05$) for each state (the first or 50% or 100% plants).

that of their female Nannong 1138-2 (48.7%). The decline of the hybrid line G9 (21.4%) was the smallest, and showed significant difference with both parents (Nannong 1138-2 and 3,060) (Fig. 5A). With regard to the dry matter accumulation (expressed as the percentage), the true hybrid lines and their parents all exhibited obvious drop under NaCl stress. Thereinto, hybrid lines G9 and G13 displayed the least drop (16.2% and 18.2%, respectively), and showed no significant difference with both parents (Nannong 1138-2 and 3,060) (18.4% and 23.0%, respectively). As followed for the hybrid line G3 (drop of 27.1%), its dry matter accumulation under salt stress was slightly lower than the two parents (Fig. 5B).

## DISCUSSION

Determining genetic purity of the hybrids is crucial to ensure reproducible breeding programs. The purity of the hybrid crops can be identified via traditional phenotypic character or through modern molecular marker-assisted selection. The former, which is relied on the different phenotypic traits of hybrids and their parental lines, is considerably restricted for the limited intuitive agronomical traits, or because some agronomical traits can be detected only in the particular plant growth period, and some traits are very vulnerable to the environmental conditions. The assessment of genetic purity of hybrids via molecular fingerprinting or banding patterns of their parental lines based on SSR markers have been utilized in some crops, such as in maize (*Hipi et al., 2013*), rice (*Bora et al., 2016*), and barley (*Ben Romdhane et al., 2018*). In our present study, the similar

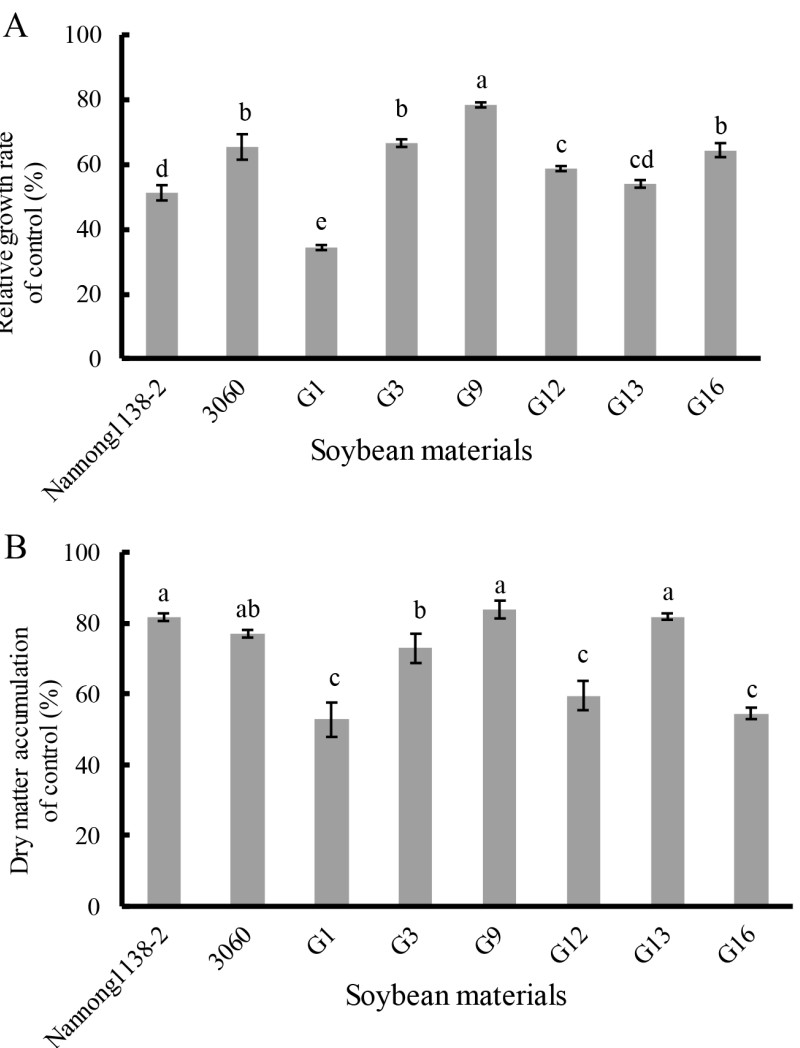

Figure 5 Changes in the relative growth rate (A) and dry matter accumulation (B) of hybrid seedlings of cross combination G and their parents under 120 mmol/L NaCl for 10 days. Note: The different lower case letters indicate significance ($p < 0.05$).

banding patterns (Fig. 3) as *Ben Romdhane et al. (2018)* were observed in the true hybrids, which are co-dominant SSR markers from both parents. However, the banding patterns of the false hybrids were only displayed in either female parent (P$_1$) (Fig. 3A, the No. 21 plant) or male parent (P2) (Fig. 3A, the No. 28 plant), even no parental bands appeared (Fig. 3A, the No. 2 plant). The false hybrids may be due to unsuccessful hybridization, impure male parent and/or under an incompletely isolated pollination resulting in no parental pollen infiltration, and so on (*Ben Romdhane et al., 2018*).Thus, 30 true hybrids (accounting for 14.1%) were successfully obtained from the harvested 213 single hybrid plants of the designed 11 cross combinations, but a large difference of the proportion of true hybrid was exhibited in different cross combinations. In cross combinations A, B, C, G, and K, the proportions of true hybrid were all more than 20%, but zero was showed in cross combinations D, E, F, and J (Table 2). For example, there were seven true hybrids (C2, C3, C9, C10, C12, C13, and C14) in the combination C with

a relatively higher proportion (accounting for 36.8% of its hybrids), following by cross combination G with six true hybrids (G1, G3, G9, G12, G13, and G16 lines, accounting for 30.0%) (Table 2; Figs. 3C and 3G).

The seedlings salt tolerant coefficient is often used as an important parameter to evaluate the salt tolerance of soybean, higher value indicating stronger salt tolerance (Li et al., 2012). Our results indicate that the salt tolerant coefficient of hybrid lines G3 and G9 was roughly equal to their male parent 3,060 at three different salt injury levels, and thus displayed their stronger salt tolerance (Fig. 4). With regard to the dry matter accumulation, hybrid lines G9 and G13 displayed the least drop (16.2% and 18.2%, respectively), but showed no significant difference with the male parent 3,060 and female parent Nannong 1138-2 (23.0% and 18.4%, respectively). Incorporation of salt tolerant genes into elite material can be done with traditional hybridization in crosses between *G. max* cultivars and *G. soja* accessions or among *G. max* accessions, and if combined with the modern molecular breeding techniques, such as the molecular SSR markers, the true hybrids with excellent traits (e.g., salt tolerance) will be possible to obtain in a shorter time (Lenis et al., 2011; Patil et al., 2016), and it will greatly benefit soybean breeders in the development of salt-tolerant cultivars.

## CONCLUSIONS

In this study, the true or false soybean hybrids of backcrosses or three-way crosses of *G. max* and *G. soja* could be identified rapidly with the parental co-dominant SSR markers at the molecular level, and 30 true hybrids (accounting for 14.1%) were obtained among 213 single plants of 11 cross combinations. The salt tolerance of six true hybrids (G1, G3, G9, G12, G13, and G16) of cross combination G and their parents was evaluated with parameters including the salt tolerance coefficient, relative growth rate and dry matter accumulation, of them hybrids G9 displayed stronger salt tolerance, and could be further used as the excellent materials for breeding new salt tolerant soybean cultivars. Thus, SSR marker technology can be a powerful tool for identifying true hybrids of soybean and, together with the simple assay of salt tolerance parameters, it can be of great significance in breeding of new salt tolerant soybean cultivars.

### Funding

This work was supported by the National Natural Science Foundation of China (No. 31671604, 30871462) and the Transgenic Engineering Crops Breeding Special Funds of China (No. 2009ZX08004-008B). The funders had no role in study design, data collection and analysis, decision to publish, or preparation of the manuscript.

### Grant Disclosures

The following grant information was disclosed by the authors:
National Natural Science Foundation of China: 31671604, 30871462.
Transgenic Engineering Crops Breeding Special Funds of China: 2009ZX08004-008B.

## Competing Interests

The authors declare that they have no competing interests.

## Author Contributions

- Fayuan Li performed the experiments, analyzed the data, prepared figures and/or tables, authored or reviewed drafts of the paper.
- Xun Liu performed the experiments, analyzed the data, prepared figures and/or tables, authored or reviewed drafts of the paper.
- Shengyan Wu performed the experiments, analyzed the data.
- Qingyun Luo analyzed the data, contributed reagents/materials/analysis tools, prepared figures and/or tables.
- Bingjun Yu conceived and designed the experiments, contributed reagents/materials/ analysis tools, authored or reviewed drafts of the paper, approved the final draft.

## Data Availability

Raw data are available as Supplemental Files.

## Supplemental Information

Supplemental information for this article can be found online at http://dx.doi.org/10.7717/ peerj.6483#supplemental-information.

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
