# Peer review of "Hybrid identification for Glycine max and Glycine soja with SSR markers and analysis of salt tolerance"

_PeerJ, doi:10.7717/peerj.6483_

## Round 0.1 · original submission · Major Revisions

Please address all the critical issues raised by all reviewers. Please also note that one of the reviewers indicated that significant English editing is needed.

Reviewer 1 ·

Basic reporting

This manuscript focusses on identifying true hybrids from backcrosses of G.max and G. soja using molecular tools such as SSR markers. Salinity is a major obstacle in crop production, causing soil degradation or erosion and urgent attention is required. Authors have chosen molecular approaches to identify and propagate true hybrids which can sustain salinity at the same time increased yield. Authors used backcrosses or three-way crosses between two distinct yet relative crops and resulting hybrids were verified for polymorphism using SSR markers. Furthermore, authors evaluated salt tolerance, relative growth, and dry matter accumulation of particular hybrids and shown that their methods created true hybrids that are salt tolerant soybean germplasm. Manuscript was well written with professional language and literature work is thorough. Introduction part was very well written with potential problems of salinity and how this can be prevented. Authors need to check for sentence formation where they need to increase scientific language rather than just normal English. Tables are represented well however figures of electrophoresis looks unclear especially Figure 2 and Figure 3 B. Authors should mark the observed bands or either mark the ladder.

Experimental design

Manuscript meets the aims and scope of the journal and experimental designs are well presented. Authors are well versed in this field and they have used proper experiments to answer key questions. Authors approaches for identifying true hybrids using backcrosses or three-way crosses and SSR markers is not novel but many groups have used this particular technique and shown to be successful. Authors done an excellent job in providing details of treatment of strains for hybridization. PCR amplification, SSR-PCR reaction system to screen polymorphic patterns among parents of different cross combinations were well designed. Authors should give the list of primers in a table so that this can be helpful for future groups if they want to test them. Authors need to mention why they have chosen true hybrid group G for salt tolerant tests after molecular appraisal results. Why not C group that have more percentage of true hybrid. Salt tolerant and dry matter accumulation experimental methods are well designed.

Validity of the findings

Results part was written well and study shows original results from designed experiments but authors need to improve discussion part as they confirm the results but not giving detailed discussion. Results from DNA extraction and true hybrid identification was well explained. Authors mentioned in Line 172 that they have chosen combination 7 as the best one. On what basis? Did they selected based on banding pattern? True hybrid selection results were well represented in the table but the explanation is kind of confusing from line 184 till 186. Selection of true hybrid based on banding pattern is interesting. Salt tolerance experiment results are slightly over interpreted. Authors mentioned in line 202 that G3 and G9 are roughly equal to their male parent 3060. However It looks like G9 is having higher salt tolerant coefficient than 3060 or G3 especially 1st and 100% plants. Dry matter accumulation results clearly show that G9 and G13 has the least drop compared to others. Conclusion part can be elaborated giving details of future work.

Additional comments

Authors have focused on identifying true hybrids from backcrosses of two different soybean crops using molecular tools such as complementary SSR markers. Authors main goal is to identify the true hybrid and further evaluate salt tolerant coefficient to see if the plants can overcome salinity at the same time maintaining proper growth rate. Authors have done excellent job in designing proper experiments for backcrosses or three-way crosses. Experimental procedures and results section was written well. However authors need to improve the discussion part in terms of results interpretation. Overall the manuscript was well written with minor revisions required. Please see some minor comments below.
Authors need to improve on sentence formation and also need to improve scientific language as some times very lament terms were used in the explaining the results.

Line 223-234 belongs to introduction part rather than discussion part.

Authors need to re write the whole sentence from line 139 to 144. Looks like there is issue in sentence formation and can’t understand what they are talking about.

Please remove So from line 252. What does it mean by “this work can be done more excellently” from Line 256. Please rephrase this sentence.

Authors need to re write the sentence from Line 265-270. Authors can mention that SSR marker technology can be a powerful tool for identifying true hybrids of soybean and this method can be of great significance in breeding of new salt tolerant soybean cultivars. Please consider this as minor comment.

·

Basic reporting

no comment

Experimental design

no comment

Validity of the findings

no comment

Additional comments

In this study the authors explored the combination of SSR markers and physiological index to identify soybean salt tolerant germ plasm. This is an interesting and thorough paper, although, its logic is largely based on previous studies (identification of soybean varieties, genetic map construction and genetic evolutionary relationship between species etc). SSR markers have also been used for variety identification and/or hybrid purity determination in rice, wheat, common bean, cotton, rape, peanut, apple, orange and psidium (Gaitán-Solís et al., 2002, Nandakumar et al., 2004; Dawson et al.,2013; Tuler et al.,2015 ). The authors established a straightforward soybean SSR-PCR system to determine parental co-dominants for hybrid identification. Further screening for salt tolerance, growth rate and dry matter accumulation resulted in identification of true hybrids while avoiding false positives. The results provided by the authors are consistent with their claims and can be published as is.

Reviewer 3 ·

Basic reporting

- The manuscript is not clear and unambiguous and it needs a professional English editing service.

- Background is sufficiently provided, but authors should stick to high-profile journal references

- There is an excess of tables (table 4 is the only important table), but figures are OK.

- Authors need to better clarifiy crosses; a figure could help on this. Authors also should consider if it is really necessary to detail all field trials before obtaining hybrids.

Experimental design

- The research question is relevant and meaningful, but authors were too wordy to have a well defined research question.

- Methods are detalied, but the section "Plant materials and treatments" is too long and confuse. Authors, based in articles that describe breeding experiments, should make a shorter explanation to explain how they obtained hybrids. In a separate section, they can explain salinity tests. DNA extraction do not have a key reference.

Validity of the findings

Authors should be more concise to explain the novelty of their findings.

---

## Round 0.2 · accepted · Accept

Critical issues pointed by the reviewers were adequately addressed and the manuscript was amended accordingly.

# Reviewer 1 ·

Basic reporting

Authors have focused on identifying true hybrids from G.max and G.soja using SSR markers. Authors have made considerable changes and improvements from their previous manuscript. Authors have also improved the scientific language with updated literature work. The figures were carefully marked with explanation in the text. Discussion part has been substantially improved. The use of SSR makers to identify hybrids and screening test for salt tolerance, growth rate and dry matter accumulation has significant advantage to improve salinity resistant crops. Overall the manuscript looks polished with clear discussion.

Experimental design

Authors done excellent job in designing suitable methods for back crosses, identification of true hybrids using SSR markers. Experiments to test the hybrids for salt tolerance, dry weight accumulation and growth rate were well designed. Authors have explained well in the response to comments for this section and I am satisfied with the responses given especially for primers and the reason for using G group but not C group as true hybrids.
Authors did not made any new experiments from previous manuscript, so new comments.

Validity of the findings

As mentioned in my previous review comments, results part was written well and study shows original results from designed experiments but authors need to improve discussion part as they confirm the results but not giving detailed discussion. Authors have done excellent job in responses for comments in this section. I appreciate your changes in the true hybrid identification of different soybeans section from results. Appreciate the changes made to the manuscript and explanation and clarification of question related to G3 and G9 differences. Acknowledge the authors for revising the conclusion part and adding details for the future research.

Additional comments

Firstly, an excellent job by the authors on carefully revising the manuscript and giving feedback to the question that were addressed from the previous manuscript. The figures look much better from previous version with markings. Authors have also made significant improvement in scientific writing and giving new references. Overall the manuscript looks good and is worth published. Thank you for the comments and clarification from previous questions.

Reviewer 3 ·

Basic reporting

Authors made an improvement in professional English along the manuscript. References are now better connected with high profile journals. The structured was ameliorated; however, the abstract (first 5 lines) goes directly to crosses; it should better describe the importance of obtaining hybrids for salinity tolerance.

Experimental design

Methods in this version are better described.

Validity of the findings

Conclusions were better stated.